# RNA Editing Alters miRNA Function in Chronic Lymphocytic Leukemia

**DOI:** 10.3390/cancers12051159

**Published:** 2020-05-05

**Authors:** Franz J. Gassner, Nadja Zaborsky, Daniel Feldbacher, Richard Greil, Roland Geisberger

**Affiliations:** 1Department of Internal Medicine III with Haematology, Medical Oncology, Haemostaseology, Infectiology and Rheumatology, Oncologic Center, Salzburg Cancer Research Institute—Laboratory for Immunological and Molecular Cancer Research (SCRI-LIMCR), Paracelsus Medical University, Cancer Cluster Salzburg, Müllner Hauptstrasse 48, 5020 Salzburg, Austria; f.gassner@salk.at (F.J.G.); n.zaborsky@salk.at (N.Z.); daniel.feldbacher@stud.sbg.ac.at (D.F.); r.greil@salk.at (R.G.); 2Department of Biosciences, University of Salzburg, Hellbrunner Strasse, 34, 5020 Salzburg, Austria

**Keywords:** chronic lymphocytic leukemia (CLL), miRNA, editing, ADAR, AID/APOBEC

## Abstract

Chronic lymphocytic leukemia (CLL) is a high incidence B cell leukemia with a highly variable clinical course, leading to survival times ranging from months to several decades. MicroRNAs (miRNAs) are small non-coding RNAs that regulate the expression levels of genes by binding to the untranslated regions of transcripts. Although miRNAs have been previously shown to play a crucial role in CLL development, progression and treatment resistance, their further processing and diversification by RNA editing (specifically adenosine to inosine or cytosine to uracil deamination) has not been addressed so far. In this study, we analyzed next generation sequencing data to provide a detailed map of adenosine to inosine and cytosine to uracil changes in miRNAs from CLL and normal B cells. Our results reveal that in addition to a CLL-specific expression pattern, there is also specific RNA editing of many miRNAs, particularly miR-3157 and miR-6503, in CLL. Our data draw further light on how miRNAs and miRNA editing might be implicated in the pathogenesis of the disease.

## 1. Introduction

Chronic lymphocytic leukemia (CLL) is a common B cell tumor of the elderly with a highly variable clinical course [1]. Patients are classically categorized according to the immunoglobulin heavy chain variable region (IGHV) mutation status of their CLL cells, with unmutated IGHV indicating a worse prognosis [2]. A broad panel of genetic aberrations have been defined for CLL, including chromosomal deletions del13q, del11q, del17p and trisomy 12 and non-synonymous mutations in more than 50 cancer drivers such as *NOTCH1*, *MYD88*, *TP53*, *ATM* and *SF3B1* [3,4,5]. While deletions del11q and del17p result in the loss of tumor suppressors ATM and P53, del13q leads to the deletion of the microRNAs miR15a and miR16-1, which regulate BCL2 expression—important for cell cycle regulation and apoptosis [6,7]. Further studies revealed that microRNAs (miRNAs) are substantially deregulated in CLL and that their specific expression patterns contribute to the development and progression of CLL and have prognostic and predictive relevance [8,9,10,11].

RNA editing is a posttranscriptional mechanism conserved in metazoans, which comprises adenosine (A) to inosine (I) deamination in RNA by ADARs (adenosine deaminases that act on RNA) [12] and cytosine (C) to uracil (U) deamination by APOBEC (Apolipoprotein B mRNA Editing Catalytic Polypeptide-like) enzymes [13,14]. As I is a guanosine analog, A to I editing has the same effect as an A to G mutation. Consequently, A to I or C to U editing can alter the protein sequence of genes, the stability of RNAs and the target sequence of miRNAs. ADAR-mediated RNA editing has been recently shown to significantly contribute to tissue specific epitranscriptome diversity in humans, by introducing A to I changes in many mRNA target transcripts [15]. In addition, many cancer cells were shown to exhibit a specific editing pattern compared to their normal counterpart cells [12,16,17,18]. From this panel of edited mRNA transcripts, many of them result in amino acid changes on the protein level with reported functional differences, contributing to proliferation and metastasis in cancer [19,20,21,22]. In this study, we analyzed miRNA editing in CLL and normal B cells. Our data revealed that, in addition to the substantial deregulation of specific miRNAs between CLL and normal B cells, CLL-specific miRNA editing occurs. Our data show, for the first time, a comprehensive analysis of miRNA editing in CLL and provide a novel insight into how aberrant miRNA expression and editing contribute to CLL pathogenesis.

## 2. Results

### 2.1. Identification of miRNAs Differentially Expressed between CLL and Normal B Cells

First, we analyzed miRNA expression in a set of 44 previously untreated CLL patients and compared it with that from 23 B cell samples. miRNAs were isolated from purified CLL cells obtained from patients enrolled in a previously reported clinical trial using rituximab in combination with fludarabine and lenalidomide (AGMT-REVLIRIT trial, ClinicalTrials.gov Identifier: NCT00738829 and 94 NCT01703364, [23]). The patient characteristics are shown in Table 1.

High throughput sequencing of microRNAs (miRNAs) was performed on the Illumina platform (Illumina, Inc., San Diego, CA, USA), and miRNA-seq data from normal B cells were accessed from repositories. By applying a threshold of false discovery rate adjusted *p*-values < 0.01, we found a total of 227 miRNAs differentially expressed between normal and malignant B cells (Figure 1, Appendix A). Although this is far more than previously reported, we could confirm the CLL-specific aberrant expression of many miRNAs, such as miR-101, miR-10b, miR-140, miR-148, miR-155, miR-181 and miR-19a [24,25]. In addition, we found four miRNAs differentially expressed according to IGHV mutation status in CLL, which were miR-125a, miR-30a, miR-99b and miR-10a (Appendix A).

### 2.2. Identification of miRNAs Editing in CLL and Normal B Cells

To determine miRNA A to I editing in CLL and normal B cell samples, we bioinformatically screened the miRNA-seq data for A to G (corresponding to I) and C to U changes. In total, we found C to U editing occurring in 11 (3.5%) and A to I editing in 14 (4.4%) of the 315 miRNAs expressed. Most of these editing events were at very low editing frequencies (<10%) and low incidence and were not restricted to CLL samples but occurred also in B cells (Appendix A). Regarding C to U editing, we found robust editing (>10% editing frequency) of miR-31 in a single CLL sample and recurrent miR-184 editing in CLL (9 of 44) as well as in B cells (5 of 23) (Figure 2A). In addition, we noticed the recurrent C to U editing of miR-106b exclusively in B cells (9 of 23) (Figure 2A). We could not observe any apparent correlation of C to U editing with miRNA expression (Figure 2B).

For A to I editing, we also found many editing events at frequencies <10% (Appendix A). However, we observed the robust (>10% editing frequency) and recurrent editing of miR-589 in two CLL samples and one B cell sample, of miR-3157 in two CLL samples and of miR-6503 exclusively in CLL cells (7 of 45) (Figure 3A). Particularly, for miR-6503, we found that editing occurred in samples with high miR-6503 expression (Figure 3B). Notably, six of the seven patients with editing of miR-6503 were IGHV-mutated (Figure 3).

### 2.3. Clinical and Biological Relevance of miRNA Editing in CLL

Next, we tested the relevance of recurrent (editing in at least two CLL samples) and robust (>10% editing frequency) miRNA editing, which we observed for miR-184 (C to U), miR-589, miR-3157 and miR-6503 (A to I). For miR-589 and miR-6503, editing occurred within the seed sequence of the miRNAs (bases 2–8 from the 5′ end of the mature miRNA), which is essential for proper binding to mRNAs [26] (Figure 4A). Hence, we next performed genome-wide target prediction for the edited miRNAs using DIANA-tools and extracted the high-confidence target transcripts (prediction score > 0.9). Thereby, we confirmed for all miRNAs different targets for their edited and non-edited versions. For miR-184, the edited version was predicted to target fewer transcripts than the non-edited miRNA (seven shared targets and two targets for the non-edited version). For miR-589, only 10 targets were shared, while 229 and 342 targets were unique for the edited and non-edited versions, respectively. For miR-3157, the edited miRNA was predicted to target two novel transcripts in addition to the 18 shared transcripts; for miR-6503, no targets were shared between the edited and non-edited versions; and 12 versus 32 unique targets are predicted to be recognized by the edited versus non-edited version of miR-6503 (Figure 4A, Appendix A).

Finally, we monitored whether miRNA editing would correlate with a specific disease development in the patients within our small cohort of 44 patients. Generally, we found that patients that exhibited editing of any miRNA had slightly more unfavorable chromosomal aberrations (Appendix A). We noticed that for patients with edited miR-184, progression free survival (PFS) was slightly longer (Figure 4B). Furthermore, patients exhibiting miR-3157 editing had a shorter (PFS) compared to patients without miR-3157 editing (Figure 4C). The characteristics of patients with miR-184 or miR-3157 editing are summarized in Appendix A. Strikingly, in multivariate analysis, miR-3157 editing remained the most powerful independent parameter compared to IGHV mutation status and the presence of the chromosomal aberrations del11q or del17p (hazard ratio: 10; 95% confidence interval: 1.8–55; *p*-value: 0.0076; Appendix A). For the other miRNAs, we could not find apparent impact on PFS or time to treatment (Appendix A).

## 3. Discussion

RNA editing substantially affects epitranscriptome diversity in humans. Apart from the recoding of mRNAs, RNA editing contributes to substantial editing of transcribed non-coding *Alu*-repeats [16,27]. This is important, as non-edited *Alu*-repeats are recognized by dsRNA sensors such as MDA5 in the cell, which elicit a pro-inflammatory IFN-response, leading to embryonic lethality [28]. However, increased IFN signaling upon ADAR inhibition in cancer seemed an attractive therapeutic option, as this could overcome immune silencing and potentiate immune checkpoint therapies [29].

Aside from in mRNA and transcribed *Alu*-repeats, editing can also occur in miRNAs and their precursors. This can alter the specificity of the miRNAs for particular target transcripts and also affect their processing to mature miRNAs and, hence, their abundance and stability [30].

Normally, the primary miRNA transcript (pri-miRNA) is processed by Drosha in the nucleus, which yields a 50–70 nt RNA loop called the precursor miRNA (pre-miRNA). The pre-miRNA is exported from the nucleus and further processed to the mature 21–23 nt dsRNAs by DICER [31]. As ADARs prefer dsRNA structures as targets, miRNAs and their precursors would be ideal ADAR substrates. Indeed, studies in *Caenorhabditis elegans* showed that more than 40% of miRNAs’ levels were altered in ADAR mutant strains, mostly reflected by increased miRNA levels and corresponding decreased pri-miRNA levels [32]. In CLL, the binding of ADARB1 to pri-miR-15/16/ impeded their further processing and resulted in the downregulation of mature miR-15/16. As miR-15/16 belong to the group of tumor suppressor miRNAs, their downregulation in leukemia likely contributes to disease pathogenesis [33].

Shoshan and coworkers showed that miRNA editing can alter their target specificity and redirects them to a different mRNA network. They showed that the wild-type, but not edited, version of miR-455 promotes the growth and metastasis of melanoma in vivo by regulating a different set of mRNAs [34]. In line with this initial report, other miRNAs were recently shown to alter target specificity in normal and malignant tissues upon RNA editing [35,36,37,38,39].

RNA editing is mediated either by ADARs (A to I editing) or by AID (activation induced deaminase)/APOBEC enzymes (C to U editing). Previous RNA sequencing studies showed that both catalytically active ADAR members (ADAR and ADARB1) as well as at least some of the AID/APOBEC family members are expressed in CLL and normal B cells [40]. However, editing is not only dependent on the presence of specific deaminases but also relies on yet poorly defined editing cofactors, which likely account for the observed differences in miRNA editing described in our study [15]. While editing by ADARs is well characterized, editing by AID/APOBECs is still more enigmatic. From the catalytically active family members (AID, APOBEC1 and APOBEC3A-H), evidence for RNA editing capabilities could so far be only shown for APOBEC1, APOBEC3A and APOBEC3G [13,14,41,42,43,44]. The editing of miR-184, described in this report, would well fit to the previously described APOBEC1 target motif A/UCA/U [14]; however, APOBEC1 is hardly expressed in CLL [40]. By contrast, APOBEC3A and APOBEC3G are both present in CLL and normal B cells, and there is also evidence for APOBEC3-mediated C to T conversions in genomic DNA from CLL cells, showing that APOBECs likely contribute to off-target DNA mutations in CLL [40,45,46].

In our study, we provide the first evidence that miRNA editing also occurs in CLL and contributes to deregulated mRNA network targeting by edited miRNAs, in addition to the previously reported editing-based alteration of miRNA expression levels in CLL [33]. Particularly, the recurrent (*n* ≥ 2) editing of hsa-miR-3157 and hsa-miR-6503 was restricted to CLL cells, which leads to many different predicted mRNA targets. Particularly, the editing of hsa-miR-3157, although only occurring in two out of 44 samples led to a shortened PFS, which should be validated and further investigated in larger cohorts.

## 4. Materials and Methods

### 4.1. Patients

Peripheral blood from 44 chemo-naïve CLL patients participating in a previously reported clinical trial (AGMT-REVLIRIT trial, ClinicalTrials.gov Identifier: NCT00738829 and NCT01703364) [23] receiving first line treatment with lenalidomide in combination with fludarabine and rituximab was collected upon informed consent and ethical approval by the Ethics Committee of the Province of Salzburg (415-E/1287/4–2011, 415-E/1287/8–2011). Sampling was performed prior to treatment starting, and CLL cells were obtained by density gradient centrifugation and a B-CLL Cell Isolation kit (Miltenyi Biotec, Bergisch Gladbach, Germany). The cell purity was >90% in all samples. The determination of prognostic markers was performed routinely in our department as described previously [23].

### 4.2. miRNA Sequencing and Bioinformatics

miRNA purification from total RNA, library preparation and sequencing on the Illumina HiSeq 2000/2500 instrument with 1 × 50 bp single reads was performed at Eurofins Genomics (Ebersberg, Germany). Demultiplexed fastq files were processed using the miARmaSeq software version 1.7 (http://miarmaseq.idoproteins.com). In brief, adapter trimming (TGGAATTCTCGGGTGCCAAGGAACTCCAGTCACCGATGTATCT) was performed with CutAdapt [47], sequences were aligned to the hg38 reference genome using STAR aligner v2.7 [48], and read counts were calculated with featureCounts [49] using gencode v31 miRNA annotations as the target list. Differential miRNA expression was calculated using the R package “edgeR” [50] with default miARmaSeq parameters (filter = yes, fc_threshold = 1).

miRNA sequencing data from normal B cells were accessed at the Sequence Read Archive (SRA) (accession number, PRJNA429049). miRNA sequencing data from CLL samples were uploaded to SRA (submission number, SUB6956031).

### 4.3. RNA Editing Analysis and Target Gene Prediction

The detection of A to I and C to T editing events was performed using the published pipeline of Alon et al. [51] (detailed at www.tau.ac.il/~elieis/miR_editing/). In brief, adapter-removed fastq files from the miARmaSeq analysis were aligned to the hg38 reference genome using bowtie1 [52] (bowtie -n 1 -e 50 -a -m 1 --best --strata --trim3 2). Aligned reads were transformed to counts of each of the four possible nucleotides at each position along the pre-miRNA sequence, for all pre-miRNAs, using 30 as the minimum quality score allowed. Next, binomial statistics were performed in order to separate sequencing errors from statistically significant modifications. Finally, known SNPs were filtered from the statistically significant modifications by manually examining the sites obtained in the UCSC genome browser.

DIANA tools MR-microT target prediction for custom miRNAs [53] (http://diana.imis.athena-innovation.gr/DianaTools/index.php?r=mrmicrot/index) was used to detect the targets of edited and unedited mature miRNA sequences of significantly modified miRNAs. Targets with a score of 0.9 or higher were considered.

## 5. Conclusions

Our study shows the first thorough miRNA-editing analysis in CLL and normal B cells. As a main finding, we show that two miRNAs (hsa-miR-3157 and hsa-miR-6503) are edited within the seed region in a subset of CLL samples but not in B cells. These editing events alter miRNA target specificity and likely affect the pathogenesis of the disease.

## Figures and Tables

**Figure 1 cancers-12-01159-f001:**
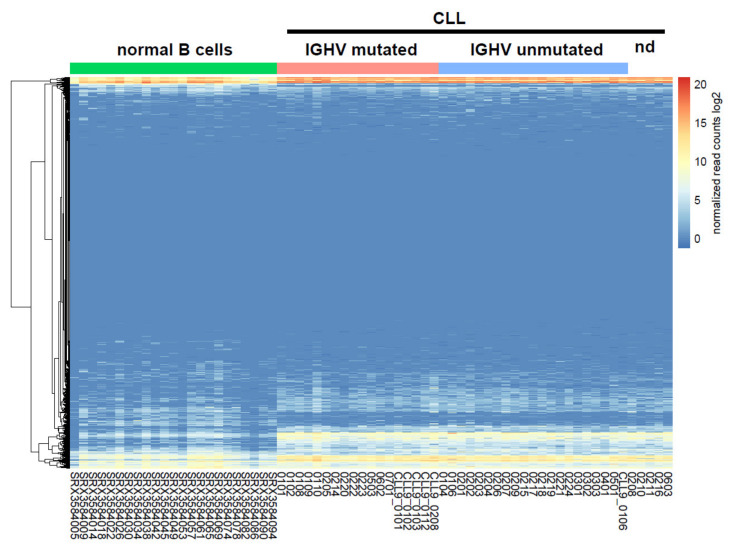
Gene expression profiles of miRNAs in CLL (*n* = 44) and normal B cells (*n* = 23). The heatmap indicates the expression values of individual miRNAs (listed on the *y*-axis) in the respective samples (listed on the *x*-axis). Significantly differentially expressed miRNAs are summarized in Appendix A.

**Figure 2 cancers-12-01159-f002:**
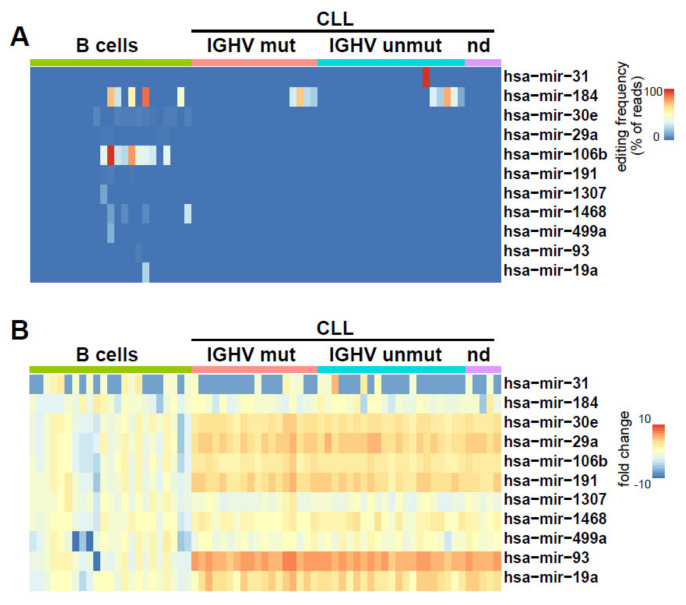
C to U editing in miRNAs from CLL and normal B cells. (**A**) Editing frequencies are indicated as a heat map in CLL (*n* = 44) and normal B cells (*n* = 23). (**B**) The heat map shows the fold change in miRNA expression in normal B cells versus CLL cells, normalized to mean expression in normal B cells.

**Figure 3 cancers-12-01159-f003:**
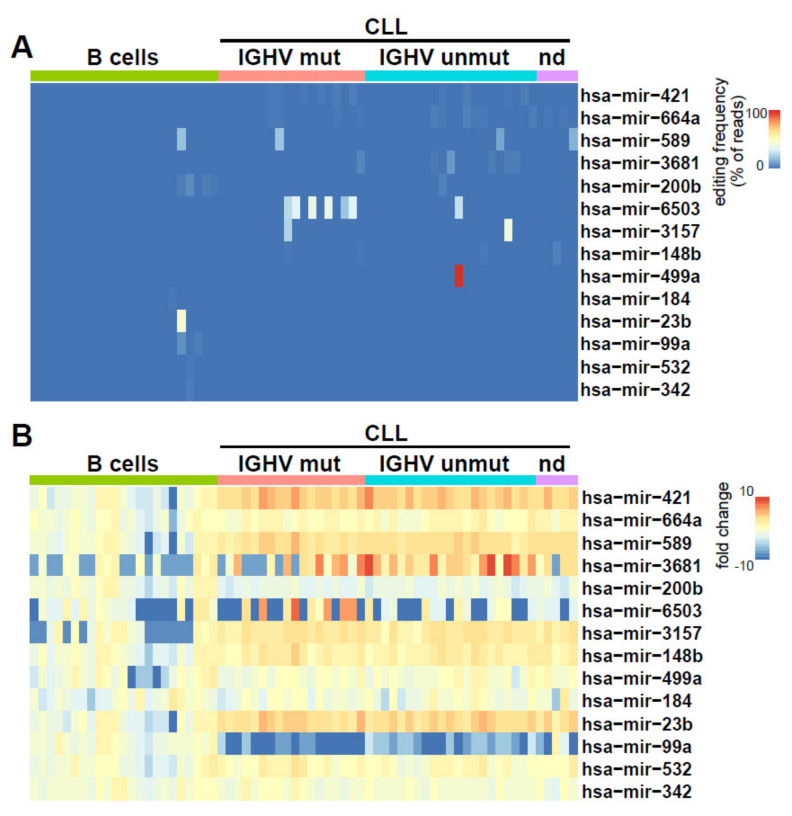
A to I editing in miRNAs from CLL and normal B cells. (**A**) Editing frequencies are indicated as a heat map in CLL (*n* = 44) and normal B cells (*n* = 23). (**B**) The heat map shows the fold change in expression in normal B cells versus CLL cells, normalized to mean expression in normal B cells.

**Figure 4 cancers-12-01159-f004:**
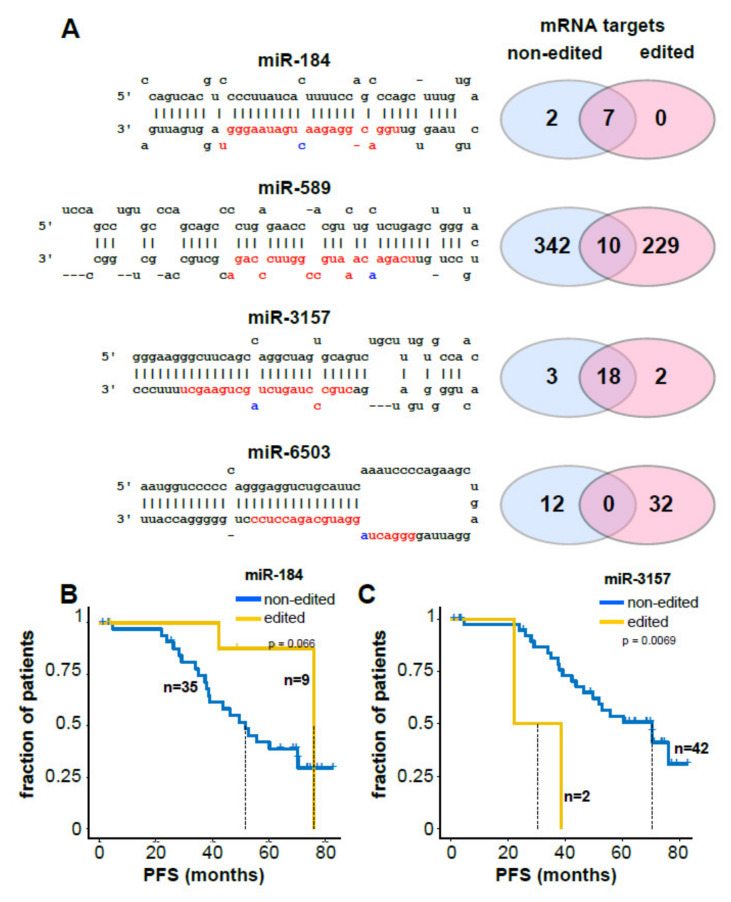
Significance of miRNA editing in CLL. (**A**) Sequences of edited miRNAs are shown. The mature miRNA is shown in red with the edited base indicated in blue. The numbers of predicted high-confidence mRNA targets for edited and non-edited miRNAs are indicated by Venn diagrams. (**B**) Progression free survival (PFS) of 44 CLL patients with or without edited miRNA-184. (**C**) Progression free survival (PFS) of 44 CLL patients with or without edited miRNA-3157.

**Table 1 cancers-12-01159-t001:** Patient characteristics.

Parameters	Total Numbers (%)
CLL samples	44 (100)
Male (%)	24 (55)
Female (%)	20 (45)
Age (years)	
Mean	65.9
Median	66.9
Range	43.3–79.8
Duration of disease (years)	
Mean	3.8
Range	0–10.3
RAI stage at diagnosis	
nda	2 (5)
I	7 (16)
II	15 (34)
III	12 (27)
IV	8 (18)
Molecular risk parameters	
Unmutated Ig VH	21 (48)
IGHV nda	5 (11)
FISH karyotype	
del11q	9 (20)
del13q	19 (43)
del17p	4 (9)
Trisomy 12	6 (14)
Normal karyotype	5 (11)
Karyotype nda	1 (2)
Treatment status	
Untreated at sampling	44 (100)
Untreated at last follow up	0 (0)

CLL, chronic lymphocytic leukemia; IGHV, Immunoglobulin variable heavy chain; FISH, fluorescence in situ hybridization; nda, no data available.

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
