# Peer review of "RNA Editing Alters miRNA Function in Chronic Lymphocytic Leukemia"

_cancers, 2020, doi:10.3390/cancers12051159_

Round 1

Reviewer 1 Report

In this study, the authors first assess miRNA expression in 44 untreated CLL patients and 23 normal B cell samples. The authors found 227 differentially expressed miRNAs between normal and malignant B cells. Next, they tested if miRNAs in their samples are A-to-I or C-to-U edited. Of all miRNAs detected, only 14 (A-I) and 11 (C-U) were edited, and not specifically so in tumors. Then, the authors look at which targets are shared or unique for the most edited miRNAs and find a surprisingly low percentage of the targets similar in the edited and non-edited miRNAs. Finally, some edited miRNAs were correlated with increased or decreased progression free survival.

Overall recommendation:

Although the findings in this study are not particularly surprising or striking, I think the authors did a good job at explaining the rationale, performed a sound analysis, and did not overstate the conclusions. In all, I recommend publication of this study in cancers.

Major:

None.

Minor:

Line 53: write B cell samples instead of 23 normal B cells.

Line 75-76: please specify what percentage of the total number of expressed miRNAs are edited.

Open questions to the authors:

Can the authors comment on the miRNA editing frequency in relation to mRNA editing frequency?

Is ADAR/APOBEC expression altered in the tumor cells in which no editing differences were noted?

Reviewer 2 Report

Overall, this work suggests novel findings in an under interrogated area of leukemia research. The piece is overall well-written with small formatting and typos. The discussion is more rich in introductory material and could be better reformatted with elements of conclusions drawn from the data. The difficulty lies in such a small sample size perhaps not being representative of disease and more importantly not representing true findings. Further, without secondary validation of target changes, biological significance is underwhelming.

  • Patient ages are given as a mean (65.9) this is slightly below the average age of diagnosis. Would like to see the median or use more patient samples to raise the age.
  • Please fix alignment of parameters in Table 1
  • A supplementary table or comment on if any of the patients with the edited miRNAs also fell into the group with abnormal karyotype
  • Figure 4B&C would be improved with associated sample sizes for edited & non-edited groups. "Borderline significance” associated with Figure 4B is misleading and should be removed. Also, which are the characteristics of these patients? Does the presence of edited miR3157 retain prognostic significance in a multivariate analysis?
  • Lines 126-139 do little to contribute to the discussion and would be better served abrogated and as part of the introduction
  • Line 141-142 the authors state "The pre-miRNA is exported to the nucleus.” I believe they meant to say “The pre-miRNA is exported from the nucleus…”
  • Purely in silico study, might benefit from validation of predicted targets in vitro.

Round 2

Reviewer 2 Report

The reviewer are addressed all my comments.  I still believe that the work could have benefited from validation of predicted targets but I understand that this may not be the expertise of this team. Nonetheless the novelty of the paper remains high.